# Facilitating Factors and Opportunities for Local Food Purchases in School Meals in Spain

**DOI:** 10.3390/ijerph18042009

**Published:** 2021-02-19

**Authors:** Panmela Soares, Sandra Suárez-Mercader, Iris Comino, María Asunción Martínez-Milán, Suzi Barletto Cavalli, María Carmen Davó-Blanes

**Affiliations:** 1Nutrition Post-Graduation Programme, Nutrition Department, Universidade Federal de Santa Catarina, 88040-900 Florianópolis, Brazil; 2Public Health Research Group, University of Alicante, 03690 Alicante, Spain; suarezmercader@gmail.com (S.S.-M.); iriscomino@gmail.com (I.C.); mariasuncion.m.m@gmail.com (M.A.M.-M.); 3Nutrition Department, Universidade Federal de Santa Catarina, 88040-900 Florianópolis, Brazil; sbcavalli@gmail.com; 4Nutrition Post-Graduation Programme, Department of Community Nursing, Preventive Medicine and Public Health and History of Science, University of Alicante, 03690 Alicante, Spain; mdavo@ua.es

**Keywords:** school feeding, sustainable agriculture, food supply, public policy

## Abstract

The objective of this study is to explore the facilitating factors and opportunities that can promote the implementation of local food purchase (LFP) in Spanish school meals in the opinions of key informants (IK). A qualitative study was carried out based on in-depth interviews with 14 KI capable of influencing Spanish food policy (Representatives of consumers and/or producers, representatives of organizations that promote LFP, and representatives of the government and/or academics). They were asked about opportunities and facilitating factors for implementation of LFP. Interviews were recorded and transcribed. A qualitative content analysis was carried out with Atlas ti. The analysis of the interviews produced two categories that include factors that- in the interviewees’ opinions- can promote LFP (social fabric and policy) and three categories that bring together the factors that represent opportunities for implementation in school meal programs in Spain (the policy agenda, regional characteristics and regional context). The overlap between social and political demands were considered to be facilitating factors for LFP. Furthermore, in the opinions of KI, the presence of health and sustainability issues on the public agenda, the existence of a structured productive system and political changes represent an opportunity to implement LFP.

## 1. Introduction

The intensification of the use of technology and the liberalization of markets has generated changes in the food system related to negative environmental, social and health impacts. The consequences of this system include the loss of biodiversity, exclusion of local farmers from production processes and the homogenization of diets [1,2,3]. There has been an observed increase in the availability, accessibility and consumption of processed foods that have a negative nutritional impact on health, given their large quantities of saturated fats, trans fats, sugar and salt [4,5].

The increase in the consumption of these foods is associated with an unhealthy diet that contributes to increased body weight, and in turn, the development of cardiovascular diseases, diabetes and liver illnesses [6,7]. The prevalence of overweight and obesity in adults, adolescents and children has been increasing around the world [8], as have the incidence and prevalence of cardiovascular diseases, diabetes and non-alcoholic fatty liver disease [7,9].

Addressing the negative consequences of the globalized and industrial food system requires public policies that promote structural changes that support a healthful and sustainable food system. In this context, in recent decades, there has been increased interest of political leaders and academics in using the power of food purchases made by public institutions to promote more sustainable and healthful forms of food production and consumption. The local food purchase (LFP) model is considered an alternative for supplying food to school feeding programs that supports forms of production and consumption that are more respectful socially, environmentally, economically and in terms of health [10,11,12,13,14,15,16].

Identification of healthful and sustainable alternatives in the provision of food for school meals represents an opportunity to promote health at school [17]. Childhood is an important developmental stage, and thus the adoption of unhealthy eating habits can have negative effects on health [3,6,7,18]. The food environment is one of the determinant factors of eating behaviors [19,20]. Evidence suggests that the foods provided at school do not always conform to dietary recommendations [21,22,23] despite the increase in overweight and obesity in the child population [8].

Prior studies show that LFP can support healthier meals provided at school [12,14,16,24]. The direct purchase of food from local farmers seems to promote healthy food offerings such as vegetables, fruit and legumes, on school menus, substituting industrially processed foods.

In Spain, meal provision is an additional service found in 58.2 percent of public education centers [25] It is used by 38 percent of school children in preschool and primary schools [26]. There are national directives that aim to stimulate the provision of healthy foods in schools [27]. However, the directives are not always applied, and school menus do not always adhere to the recommendations [22,28]. Also, the regulation for public contracting supports the inclusion of environmental and social criteria in the selection of suppliers, but there is still no specific criterion that promotes the direct purchase of products from the local region for school meals. Even so, some regions are developing strategies to incorporate these foods into their school meals [15,29].

Prior studies show that the implementation of this strategy is more frequent in rural areas [29] and in regions that have government support. However, evidence shows that this support is not sufficient to implement LFP in all schools and/or regions [29,30].

Despite the benefits of locally purchasing foods, both for school meals and in terms of promoting health and sustainable food systems [10,14,30,31], not all schools engage in directly purchasing foods from local producers [15,29,30]. Knowing the facilitating factors and opportunities for LFP implementation can help support these initiatives.

The objective of this study is to explore the facilitating factors and opportunities that can promote the implementation of LFP in Spanish school meals in the opinions of key informants.

## 2. Materials and Methods

A qualitative study was carried out based on in-depth interviews with key informants (KI) with the capacity to influence food policies in Spain. Participants were selected using a list of the possible KI, identified through the Internet and using the snowball technique. The initial list was discussed among the researchers until a consensus was reached. Key informants were contacted by email and/or telephone to explain the objective of the study and invite them to participate. Those who accepted were sent information via email about the study, and a time, date and place was set to carry out an interview.

Interviews were carried out between May and June of 2015 until saturation of data was reached. Fourteen interviews were carried out, grouped into three categories based on experiences and work area: 1. Representatives of consumer and/or producer organizations (n = 4), 2. Representatives of organizations that support local food purchases for school meals (n = 5), and 3. Representatives of the government and/or academics with the capacity to influence public policy (n = 5).

Study participants were selected with different professional profiles related to the local purchase of food products. The group “producers and consumers” included national organizations related to agricultural production and food consumption in schools. The group also included a farmer with training in agronomy and experience in the implementation of a regional LFP program. The group “representatives of organizations that support LFP for school meals” included representatives of national organizations that participated in local food purchase initiatives working with the school food service. Finally, the group “government and/or academics with the capacity to influence public policy” included experts in public health, economics, health promotion, food security and rural farming with ample professional and/or research experience.

Key informants were interviewed in their workplaces, except for one informant who was interviewed in a public place. Before beginning the interview, the researcher reviewed the objectives of the study and guaranteed the anonymity of the participants, who signed a consent form based on ethical research protocols. The study was approved by the committee for ethics in research with human subjects of the University of Alicante (nº UA-2015-03-31).

A guide was used for the interviews that included 13 open-ended questions. It was prepared by the research team, with the collaboration of public health experts. Three pilot interviews were carried out that were included in the analysis and served to test and improve the interviews. The topics addressed were: 1. Opinion about local food purchase policies, 2. Spanish LFP initiatives in school feeding, and 3. Opportunities and facilitating factors for the implementation of local food purchase policies in school meals. The average interview time was 55 min. The interviews were recorded, with the consent of the interviewees, and transcribed literally. The texts were imported into the qualitative analysis software Atlas. ti, version 7.5. (Scientific Software development GmbH, Berlin, Germany).

In order to identify the opportunities and facilitating factors for implementing LFP, a qualitative content analysis was carried out. First, we carried out repeated readings of the interviews, to become familiar with their content. The analysis was carried out using an inductive and open coding process that coded fragments of the text with the same meaning. Later, inter-related codes were grouped to establish categories and subcategories. An independent analysis of peers was used to compare and contrast the resulting categories. Disagreements were brought to a third researcher for review.

## 3. Results

The analysis of the interviews resulted in five categories that bring together the factors that, in the opinions of those interviewed, could promote and/or suppose an opportunity for the implementation of the purchase of local food products for school meals. These categories were divided into 14 subcategories. The codes grouped into each category are shown in Table 1.

### 3.1. Facilitating Factors of LFP

The two categories that resulted in the facilitating factors of LFP identified by the KI were: social fabric and policy (Table 1).

#### 3.1.1. Social Fabric

The region’s social fabric, interpreted by KI as key actors, social pressure and pilot initiatives, was considered important to push for and influence the implementation of LFP. In referring to key actors, the KI identified both governmental actors and non-governmental actors. The ministries of agriculture, health and education figured among the key actors. They also highlighted the importance of education to influence and raise awareness among different groups involved in the food system (including groups involved in activities ranging from production to consumption in schools). Specifically, they pointed to the education community/consumers, producers and managers.


*“One of the things that would have the most influence is education. Education of the child who is going consume, you can teach him what’s good, what’s bad, or within what he likes; it’s a question of education of the little ones. And later, probably also an assessment, an education of the producer.”*
(Producers)


*“The point of view of the producer has never been good, even on the part of citizens. It’s common, for example, that someone stops on the side of a road, grabs a head a lettuce that’s there and takes it away in his car. It seems so natural, he doesn’t even realize it’s the effort and the work of another person. Education would really help in this way.”*
(Producers)

Among the non-governmental organizations, the influence of consumers, producers and business sector was highlighted. The coordination among all these groups was considered necessary to implement LFP.


*“If we don’t work together in a more coordinated way in terms of business, with the monitors, with the families, the teachers, the project team...If we don’t increase our level of coordination, which before was little, it will not work out. Things don’t work out because we need everyone working at the same time so that it makes sense and the kids see both the educational part and that it makes sense to eat differently.”*
(Local purchase supporters)

The power of the pressure of society is also considered to be relevant to influencing the political agenda and local agenda towards implementing LFP. As an example, they signaled pilot initiatives put into practice by different non-governmental organizations. They considered that generating evidence of the potential benefits and transferring them to public management could exercise and increase such pressure.


*“Another fundamental actor are the many consumer groups that have been created by organized citizens, from the perspective of the consumer, but they are also related to small farmers, well, they have a certain political projection. I’m not saying it to take sides, it’s that they express the need to support all of these types of things, they can organize themselves a bit more, we all can organize ourselves more in this sense.”*
(Local purchase supporters)


*“Showing with research the positive impacts that it could have (LFP), and on the other hand we also get research, a guide, so that it’s easy for public institutions to switch to a public purchase model.”*
(Local purchase supporters)

#### 3.1.2. Policy

Policy, interpreted by KI as political will, political ideology and norms, was referred to as another of the key factors for the development of LFP in school meals. In a context of political and economic decentralization, the potential of political will at the different levels of government (national, regional, local) was highlighted.


*“On the other hand, the regions also have many possibilities because they determine school policies, then, even though there are policies that depend on the central government, the regions can legislate more. They have normative capacity.”*
(Academics)


*“When the competencies are of the regions, each one does what it wants, and that is when you reach the conclusion that it depends on the political will of each government. You can reach one extreme or the other.”*
(Specialists)

However, they recognized that political will is related to the ideology of those who govern at the different levels of government.


*“In Andalucía there are also meals with local food purchases, above all in rural areas, in towns. There was a program of the prior government, and we have information that it’s disappearing because they are considering establishing catering businesses, in other words, it’s being lost.”*
(Consumers)

It is for this reason that being able to count on a national directive was considered important to promote LFP in the whole of Spanish territory, independently of the political will and ideology of the government.


*“I think that these initiatives are really related to progressive movements, of the left, and that it should be more transversal. I mean, we’re not talking about ideologies, rather it’s something that I think is important for everyone, for the whole society and for the region, independently of ideology.”*
(Producers)


*“National policy has to go in this direction.”*
(Local purchase supporters)

The regulations of the European Community were considered supporters of possible changes in the food supply strategies for school meals.


*“After having approved the European directive on public purchasing, we saw for the first time the possibility to link social and ecological criteria to purchasing”*
(Local purchase supporters)

### 3.2. Opportunities for the Development of LFP

With respect to the opportunities for the development of LFP in school meals, the analysis of the interviews showed three categories that group together the opinion of the KI: the public agenda and the characteristics and context of the region (Table 1).

#### 3.2.1. The Public Agenda

The fact that the topics of health, food and sustainability currently form part of the public agenda as well as the increasing demand for foods that are produced by more sustainable farming systems was considered an opportunity to support changes in the food supply strategies for school meals.


*“The quality of the food that is prepared on-site and purchased with proximity criteria, of course, there has been an upturn. The families have valued it, and now we’re realizing that what we were eating from businesses and well, it’s very different to eat when we consume products that are in season and are locally produced. Of course.”*
(Consumers)


*“The experience of school gardens in some schools, could have generated interest from this point of view, a little bit due to the sustainability, for incentivizing the local economy.”*
(Government representatives)

However, the right to a safe and adequate food supply was mentioned as an opportunity to promote LFP on fewer occasions. 


*“One of the things that we should support or defend is what we were talking about, the right to food, because all children have a right to proper foods at school, and therefore the guarantee of an adequate food supply each day at school shouldn’t be dependent on the purchasing power of the family, it should be guaranteed by the public administration.”*
(Specialists)

#### 3.2.2. Characteristics of the Region

The characteristics of the region, interpreted as geographical, agricultural and cultural characteristics linked to infrastructure and a sense of belonging, was another of the opportunities recognized by the majority of the KI for the implementation of LFP. Specifically, they highlighted the importance and influence of active rural communities with structure and productive capacity. The availability of foods, conditioned by the characteristics of the region, and the availability of adequate infrastructures for production, distribution and storage of foods was highlighted as an opportunity to promote LFP strategies.

In particular, these infrastructures included the existence of producer networks with which schools could have contact to facilitate food supply, technical assistance for direct sales, ecological production centers, agricultural cooperatives and purchase centers. One KI even considered that the development of new technologies could be an opportunity to make visible alternative ways of commercializing foods.


*“The characteristics of the region itself, agrofood and agricultural structures, ... we know that some places have many more than others.”*
(Local purchase supporters)


*“In Catalonia, which is the place I know best, there is a network of producers with whom it is very easy to connect, and they’re organized.”*
(Local food supporters)


*“It’s possible to coordinate better with these new tools, that is, it’s possible to have control. For example, a school could end up having control and even purchase food products from local producers using the Internet. And the local producers could also organize themselves in many ways.”*
(Producers)

The feeling of regional belonging was considered an opportunity to promote LFP, given the value the population attributes to the agriculture of the region.


*“I think that Asturians are people who very much love their land, that is, the fact that they love the land means that they value what is local; and valuing what is local makes what’s yours better.”*
(Producers)

#### 3.2.3. Context of the Region

The economic, political and agricultural context was also considered a possible opportunity for implementation of LFP, given that during economic recession LFP could contribute to provide employment and improve rural economy.


*“So that young people don’t leave, and many young people who have lost their jobs in the city come back to the towns. And if they have a school, they come back, so it’s an added value and generates the settling of the population that doesn’t go to the big cities, above all...the biggest benefit that we see.”*
(Consumers)

Furthermore, the political context was considered a window of opportunity for introducing changes in food policy. The KI signaled that the changes in government could bring about ideological changes that favor implementation of policies or initiatives that promote LFP in different levels of government.


*“There’s a change in government, political will changes and all of a sudden anything is possible.”*
(Local purchase supporters)

Finally, the need to find new markets for small producers in the region could also be considered an opportunity, given that school meals would represent a stable market that could strengthen the agricultural sector. 


*“And on the other hand, the need of farmers to find new markets, new short circuits, of access to markets.”*
(Local purchase supporters)

## 4. Discussion

The current study shows the facilitating factors and opportunities that in the opinions of KI exist for the implementation of LFP in school meals in Spain. The social fabric and policy were identified as facilitating factors of LFP, while topics including health, food and sustainability in the public agenda, and the characteristics and context of each region were identified as opportunities for the implementation and development of LFP.

Similar to what has been found in prior studies [24,32,33] he capacity of key actors and civil society’s power to influence decision makers was considered relevant for the implementation of LFP. Furthermore, political factors were also considered important. This coincides with the results of a prior study that showed the implementation of LFP can be more difficult in conservative states than in ones that are more liberal or socially democratic [34]. The will of political groups was also considered influential in terms of incorporating changes in policies related to food and nutrition [34,35]. Thus, political ideology is one of the factors that could explain why LFP does not have the same level of development in different regions of the country, given that only some regions have projects or programs related to LFP in their school meals [15]. 

Offering foods in schools that diverge from health guidelines can have a negative impact on the health of the child population and contribute to the development of nutrition-related illnesses [6,18]. The European population conforms to a pattern of unhealthy eating. As such, limiting consumption of sugar, salt and animal fats, and increasing consumption of vegetables, fruits, dairy products, eggs, fish and vegetable oils is considered necessary to reduce the risk of developing illnesses related to unhealthy diets [5]. Purchasing foods from farmers from the school’s region can promote the offer of healthy foods on school menus [12,14,16,24].

This is why LFP is considered a health promotion strategy. However, different levels of LFP development could suppose unequal access of the school population to an adequate diet. The existence of a strategic initiative at the national level could help balance out this situation as well as support LFP in all regions.

In light of the intrinsic relationship between a healthy diet and a sustainable food system, in recent years the international debate on nutrition, sustainability and health has expanded [36]. Consumers have also become more sustainable in their behavior [37]. According to our results, the presence of these issues on the public agenda was considered to be an opportunity to promote the implementation of LFP. This is because implementing LFP in schools can have a positive impact on the environment and on population health. Furthermore, it promotes the inclusion of fresh foods in school menus (such as fruit and vegetables) [14], and prior studies show that LFP favors the inclusion of food products that are produced in a way that is more respectful of the environment [29]. In addition, the regional provision of foods could support a decrease in the emission of greenhouse gases [38].

The results suggest that the political and economic context could be an opportunity for the implementation of LFP in school food services. In this study, a change in government was considered to be an opportunity to modify strategies for food provision to these services, as has happened in other countries [24]. Also, our results highlight the economic crisis as an opportunity for LFP implementation. The local purchase of food is a mechanism that contributes to creating stable markets for local agriculture [39], increasing the number of farm jobs in rural areas and reducing general unemployment during a recession period. In this sense, local purchasing could contribute to improving the socioeconomic circumstances of the rural population and consequently support their academic achievement and state of health [40,41]. Furthermore, local food purchases that supply school feeding programs can contribute to increasing the diversity of the foods produced and consumed in the region [42]. This result is especially important considering the low consumption of healthy foods such as fruits and vegetables among the population [5]. The availability of these healthy foods is not always sufficient to meet the needs of the population for a healthy diet, as shown in various studies [42,43]. Development of local food purchase policies that support local farmers can contribute to increasing the production and availability of healthy foods locally.

The agricultural characteristics of the region and the presence of adequate infrastructures for transport, storage and distribution of food products were other areas designated as opportunities for the implementation of LFP in school meals by the participants in this study. Prior studies have identified a greater predisposition towards LFP in school meals located in rural areas [29], in those which have access to nearby production [44], that have an on-site kitchen [29,44] and with adequate infrastructure [45], or a food center for the region [46]. Greater investment in infrastructure for the production, storage and distribution of food products could contribute to the implementation of LFP and generate positive impacts for local farmers (work and income) [39], for consumers (quality of school menus) [14] and for the environment (decreasing the emission of greenhouse gasses) [38].

One of the limitations of this qualitative study is that the results are based on the opinions of 14 key informants. As such, they can not be generalized. Furthermore, the opinions of those interviewed could be conditioned by their prior experience. However, the selection of participants with different professional or association-related backgrounds allowed for a first approach to the factors and opportunities that could favor the implementation of LFP in Spain. Furthermore, it must be taken into account that in the time that has elapsed since the data was collected, LFP policies have advanced in some regions. However, Spain still does not have a national strategy to promote LFP in school feeding. One of the strengths of our study is that it highlights the opportunities and factors that promote LFP in school feeding in a context of growing interest in strengthening sustainable food systems.

The direct purchase of foods from local farmers for school meals can impact different health determinants including nutrition, the economy and the environment. Given the potential benefits for health, knowledge about the opportunities and facilitating factors for implementing direct purchase policies can be useful for health policy and planning. Our results suggest that the consolidation of a national strategy to support LFP, together with greater investment in infrastructure for food production, storage and distribution in different regions, could contribute to the implementation of LFP policies.

## 5. Conclusions

The coordination between the social fabric demands and political will were considered to be the facilitating factors for LFP. The social fabric of the region, defined by the KI as the social pressure and pilot initiatives carried out, was considered important in influencing the implementation of local food purchasing. Also, ideology and political will together with the existing national and international norms related to food purchasing were considered keys for pushing for changes in strategies for supplying food to school food services.

In the opinions of the KI, the public agenda, and the political, economic and farming context of the region constitute an important opportunity for implementing LFP. Specifically, health, nutrition and sustainability are topics on the public agenda, there is a rural region with a structured capacity, and there is rural demand for employment.

The current study has laid out facilitating factors and opportunities that could promote the implementation of local food purchasing, without considering factors that could make implementation more difficult. Future research should explore implementation difficulties from the perspective of the institutions that are carrying out local food purchase programs. This information could help address the difficulties involved in consolidating healthier, more sustainable food systems.

## Figures and Tables

**Table 1 ijerph-18-02009-t001:** Categories, Subcategories and Codes for the Facilitating Factors and Opportunities for the Purchase of Local Foods in School Meals.

	Category and Subcategory		Codes
**Facilitating Factors**	**Social fabric**	**Key actors**	(public administration), (ministry of agriculture and food), (state and regional government), (economic sector), (health sector), (education sector), (consumers and their organizations), (producers and their organizations), (organizations of health professionals), (business sector), (communications media)
		**Social pressure**	(social alliances), (growth of the concept of food sovereignty), (coordination of the food system) (institutional meals), (population decisions about policies), (consumer demand), (strength of civil society); (social empowerment), (pressure of supporting initiatives), (pressure of the organizations of political parties), (social pressure)
		**Pilot initiatives**	(regional programs and projects), (political initiatives), (municipal projects) (individual school initiatives), (individual organizational initiatives)
	**Policy**	**Political will**	(political will), (political will /administrative structure), (voluntary policy/pressure of civil society), (political will/ideology), (political will/purchase criteria)
		**Political ideology**	(political ideology), (ideology/political context)
		**Regulations**	(new European framework), (food policy), (European policy), (fiscal policy), (policy/regulations), (political/economic)
**Opportunities**	**Public agenda**	**Health**	(Health policy agenda), (social demand), (importance/ concern about diet), (interest in food and health)
		**Right to food**	(right to food), (food security)
		**Sustainability**	(awareness/sustainability), (sustainable and healthy food)
	**Regional characteristics**	**Agricultural geography and culture**	(local agriculture), (productive capacity), (geographic and agricultural characteristics), (food culture), (autonomous development), (food availability), (scale of production), (Purchase traditions)
		**Infrastructure**	(technical assistance), (infrastructure), (infrastructure/initiatives), (producer network), (new technologies)
		**Sense of regional belonging**	(territorial defense), (autonomous development), (sense of autonomy)
	**Regional context**	**Economic**	(economic/political support) (change in perception about meals quality) (economic crisis/awareness raising) (crisis in Spain) (crisis/questioning the economic model) (crisis/promotion of local employment) (crisis/return to rural regions)
		**Political**	(policy change) (political change) (political context)
		**Agricultural**	(market stability) (rural exodus/promotion of the agricultural sector) (needs of farmers)

## Data Availability

The data presented in this study are available on request from the corresponding author. The data are not publicly available due to privacy.

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
