# Peer review of "Facilitating Factors and Opportunities for Local Food Purchases in School Meals in Spain"

_ijerph, 2021, doi:10.3390/ijerph18042009_

Round 1

Reviewer 1 Report

The present study by Soares et al. requires an extensive English edition. Some parts of the text are difficult to understand.

The Introduction needs to be expanded and better reasoned. For example: “Prior studies show that the LFP can support healthier meals provided at school” – This needs na justification.

The data analys needs to be better explained on section 2.

Study limitations need to be addressed at the end of Discussion.

The Conclusions need to be expanded and better justified. “The coordination between the social fabric demands and political will were consid- 303 ered to be the facilitating factors for LFP” – Why do you say this?

Future investigations on the issue should be pointed out by the authors at the end of Conclusions.

Author Response

Response: We appreciated your time in reviewing this work. In this new version, we have incorporated the suggestions of the reviewers, and we believe that this has contributed to improvements in the manuscript.

  1. The present study by Soares et al. requires an extensive English edition. Some parts of the text are difficult to understand.

Response: We have reviewed the text and made all the needed corrections. The text has been revised by an English native speaker.

  1. The Introduction needs to be expanded and better reasoned. For example: “Prior studies show that the LFP can support healthier meals provided at school” – This needs a justification.

Response: In response to the comments of the reviewer, we have included information in the introduction that justifies carrying out this study. Please see the introduction, page 2.

  1. The data analysis needs to be better explained on section 2.

Response: In this new version we have tried to better explain data analysis. Please, see the sixth paragraph of the methodology, page 3.

  1. Study limitations need to be addressed at the end of Discussion.

Response: In the first version sent, the limitations were in the last paragraph of the discussion. However, we believe that this information was not clear enough. In this new version, we have improved the wording of the limitations. Please, see the seventh paragraph of the discussion, page 09. 

  1. The Conclusions need to be expanded and better justified. “The coordination between the social fabric demands and political will were considered to be the facilitating factors for LFP” – Why do you say this?

Response: In response to the comments of the reviewer, in this new version we have provided information in the conclusions section. Please see the conclusions section, page 09.

  1. Future investigations on the issue should be pointed out by the authors at the end of Conclusions.

Response: In this new version we have included relevant directions for future investigations at the end of the conclusion. Please see the third paragraph of the conclusions section, page 9.

Reviewer 2 Report

The manuscript is interesting, the methodology used is adequate, and the results support the discussion. However, I have the following comments.

I. Major Comments:
1. The instruction is consistent with the purpose of the study, however it is necessary to include more aspects related to the nutrition and health of children.

1.1. I suggest including a brief paragraph on the components of food that are related to overweight and obesity, and diseases such as diabetes mellitus, high blood pressure and NAFLD especially in children (total fat, saturated fat, sugar, sodium, etc.).

1.2. It is necessary to include more information regarding the impact on health and nutrition, and the purchase of unhealthy food at school.

Suggested reference:
Relevant Aspects of Nutritional and Dietary Interventions in Non-Alcoholic Fatty Liver Disease. Int J Mol Sci. 2015; 16: 25168-98.
PMID: 26512643

2. In the discussion I suggest including aspects such as educational level and the presence of chronic non-communicable diseases.
Suggested references:
Overnutrition and Scholastic Achievement: Is There a Relationship? An 8-Year Follow-Up Study. Obes Facts. 2018; 11: 344-359.
PMID: 30308520

A multifactorial approach of nutritional, intellectual, brain development, cardiovascular risk, socio-economic, demographic and educational variables affecting the scholastic achievement in Chilean students: An eight-year follow-up study. PLoS One. 2019; 14 (2): e0212279.
PMID: 30785935

3. In the discussion, it is necessary to include more studies related to the impact on public health. Especially grocery shopping at school and unhealthy eating.

4. The wording of the conclusion "is confusing", I suggest rewriting the discussion.

II. Minor comments:
1. Improve the writing of the study objective.
2. I suggest including a brief paragraph in the discussion regarding the development of potential health and nutrition policies.

Author Response

The manuscript is interesting, the methodology used is adequate, and the results support the discussion. However, I have the following comments.

Response: We appreciated your time in reviewing this work. In this new version, we have incorporated the suggestions of the reviewers, and we believe that this has contributed to improvements in the manuscript.

  1. Major Comments:
  2. The instruction is consistent with the purpose of the study, however it is necessary to include more aspects related to the nutrition and health of children.

Response: In response to the reviewer's comments, we provided information in the introduction related to the nutrition and health of children. Please see the second and fourth   paragraphs of the introduction, pages 1 and 2.

1.1. I suggest including a brief paragraph on the components of food that are related to overweight and obesity, and diseases such as diabetes mellitus, high blood pressure and NAFLD especially in children (total fat, saturated fat, sugar, sodium, etc.).

Response: We have included in the introduction information regarding the components of food that are related to overweight, obesity, and chronic non-communicable diseases. See the first paragraph of the introduction, page 1.

1.2. It is necessary to include more information regarding the impact on health and nutrition, and the purchase of unhealthy food at school. Suggested reference: Relevant Aspects of Nutritional and Dietary Interventions in Non-Alcoholic Fatty Liver Disease. Int J Mol Sci. 2015; 16: 25168-98. PMID: 26512643

Response: Thank you very much for the reference. In the introduction of this new version of the manuscript, we have included information related to the health and nutrition impacts of the purchase of unhealthy food at school. See the 2º paragraph of the introduction, page 1.

  1. In the discussion I suggest including aspects such as educational level and the presence of chronic non-communicable diseases. Suggested references: Overnutrition and Scholastic Achievement: Is There a Relationship? An 8-Year Follow-Up Study. Obes Facts. 2018; 11: 344-359. PMID: 30308520 A multifactorial approach of nutritional, intellectual, brain development, cardiovascular risk, socio-economic, demographic and educational variables affecting the scholastic achievement in Chilean students: An eight-year follow-up study. PLoS One. 2019; 14 (2): e0212279. PMID: 30785935

Response: Thank for the reference. In response to the reviewer's comments, we provided information in the discussion related to the educational level and the presence of chronic non-communicable diseases. Please see the fifth paragraph of the discussion, page 8.

  1. In the discussion, it is necessary to include more studies related to the impact on public health. Especially grocery shopping at school and unhealthy eating.

Response:  In this new version, we include information related to the impact on public health of buying unhealthy food at school. Please see the third paragraph of the discussion, page 8.

  1. The wording of the conclusion "is confusing", I suggest rewriting the discussion.

Response:  We have changed the writing of the conclusion. Please see the conclusions section, page 09. 

  1. Minor comments:
  2. Improve the writing of the study objective.

Response: We have changed the writing of the objective. Please see the last paragraph of the introduction (page 2) and abstract (page 1).

  1. I suggest including a brief paragraph in the discussion regarding the development of potential health and nutrition policies.

Response: In the discussion of this new version of the manuscript we have included information on how our results can contribute to the planning of food, nutrition, and public health policies. Please, see the last paragraph of the discussion section, page 09.

Reviewer 3 Report

The issue of consumers’ behaviour in the food market has been my main scientific interest for years. In March 2020 I completed a research project entitled “Changes in food consumption models in Poland.” That is why I am very happy to have the opportunity to read an interesting article on this topic.

The article is well-written and has a research character. The authors should be appreciated for using in-depth interviews (IDI). The article also presents the views of a wide group of respondents: Representatives of consumer and/or producer organizations. Representatives of organizations that support local food purchases for school meals, and representatives of the government and/or academics with the capacity to influence public policy. However, the described study is quite old. IDI was conducted from May to June 2015 (line 68). Why did the authors take so long to publish them? What is the significance of this research today, after nearly six years?

In my opinion, before publishing the work, the Authors should make some improvements to their text:

  1. In the Introduction section, state research questions.
  2. The authors provide a comprehensive description of the research method and the process of measurement, but we do not know almost anything about people taking part in the research. A broader characterization of key informants (KI) is necessary. The current data is too general. How competent were these people?
  3. When posting excerpts from respondents' statements, please indicate clearly whose voice it is. Is it the voice of producers or consumers, specialists, academics or government representatives?
  4. In line 103, p.3, the Authors refer to Table 1. This table is presented on p.7. This situation makes it difficult for the reader to correctly perceive the text. This table should be placed just after the reference.
  5. The discussion is very poor and there is no literature review at all. Authors must refer to the results of their research to the already existing knowledge on this subject, published in the literature of the subject. I especially recommend the article: Maciejewski, G., Consumers Towards Sustainable Food Consumption, Marketing Of Scientific And Research Organizations 2020, Vol. 36, Issue 2, p. 19-30, DOI: 10.2478/minib-2020-0014, and also Maciejewski, G., The food expenditure in Poland and other European Union countries – a comparative analysis, Olsztyn Economic Journal 2019, No.14 (2), p. 179-194, DOI:10.31648/oej.3970.
  6. The Conclusions section should be expanded. Currently, it practically does not exist. The authors limit themselves to just two sentences: „The coordination between the social fabric demands and political will were considered to be the facilitating factors for LFP. Furthermore, in KI opinion, the presence of health and sustainability issues on the public agenda, the existence of a structured productive fabric and political changes represent an opportunity to implement LFP.” (line 303-306). It is not enough. What are the conclusions for producers from these studies? For consumers? For local purchase supporters? For government representatives?
  7. Moreover, in the Conclusions section, it should be clearly stated about the limitations of the method used and the study itself, which is qualitative and not representative. In the end, indicate further directions for research. Will the Authors seek confirmation of the obtained results in quantitative research?
  8. It would be better to include a research tool in the appendix.

I believe the authors will have no trouble improving their article. Good luck!

Author Response

The issue of consumers’ behaviour in the food market has been my main scientific interest for years. In March 2020 I completed a research project entitled “Changes in food consumption models in Poland.” That is why I am very happy to have the opportunity to read an interesting article on this topic.

The article is well-written and has a research character. The authors should be appreciated for using in-depth interviews (IDI). The article also presents the views of a wide group of respondents: Representatives of consumer and/or producer organizations. Representatives of organizations that support local food purchases for school meals, and representatives of the government and/or academics with the capacity to influence public policy. However, the described study is quite old. IDI was conducted from May to June 2015 (line 68). Why did the authors take so long to publish them? What is the significance of this research today, after nearly six years?

Response: We appreciate your time to review this work. The interviews were conducted in 2015 during doctoral studies. With the completion of the doctoral thesis and without funding, the project had to stop. We were able to take it up again in 2019 when one of the authors obtained a postdoctoral fellowship.

Several years have indeed passed since the data was collected. However, Spain has not yet promoted the purchase of locally produced food in large-scale school feeding services. In fact, one of the strengths of this work is to help identify promoting factors and opportunities that can contribute to the implementation of local purchasing programs in a context of growing concern for the sustainability of the food system. This information is in the study limitations paragraph. Please, see the seventh paragraph of the discussion section, page 10.

In my opinion, before publishing the work, the Authors should make some improvements to their text:

We appreciate the opportunity to review the manuscript. In this new version, we have incorporated the suggestions of the reviewers, and we believe that this has contributed to improvements in the manuscript.

  1. In the Introduction section, state research questions.

Response: In this new version we have incorporated the research questions in the introduction section. Please, see the last paragraph of the introduction section, page2.

  1. The authors provide a comprehensive description of the research method and the process of measurement, but we do not know almost anything about people taking part in the research. A broader characterization of key informants (KI) is necessary. The current data is too general. How competent were these people?

Response: In the participant selection process, we seek to identify informants with different professional experiences. In the group of producers and consumers we identified important national organizations related to agricultural production and consumption in schools. Also, a farmer was included for his training in agronomy and his experience in starting a local program of direct purchase of food from farmers in the region. In the group of Representatives of organizations that support the local purchase of food for school service, we selected representatives of national institutions that had participated in initiatives to purchase local foods in school food service. Finally, in the group of specialists, academics and government representatives, we identified experts in matters of public health, economics, health promotion, food security and rural production. The participants of this group had extensive professional and research experience. We have included this information in the methodology section. Please see the third paragraph of the methodology, page 3.

  1. When posting excerpts from respondents' statements, please indicate clearly whose voice it is. Is it the voice of producers or consumers, specialists, academics or government representatives?

Response: In this new version in the results section we indicate whose voice is in the statement extracts. Please see the results session, page 3.

  1. In line 103, p.3, the Authors refer to Table 1. This table is presented on p.7. This situation makes it difficult for the reader to correctly perceive the text. This table should be placed just after the reference.

Response: In this new version we have changed the location of the table in the text. Please, see the table on page 4.

  1. The discussion is very poor and there is no literature review at all. Authors must refer to the results of their research to the already existing knowledge on this subject, published in the literature of the subject. I especially recommend the article: Maciejewski, G., Consumers Towards Sustainable Food Consumption, Marketing Of Scientific And Research Organizations 2020, Vol. 36, Issue 2, p. 19-30, DOI: 10.2478/minib-2020-0014, and also Maciejewski, G., The food expenditure in Poland and other European Union countries – a comparative analysis, Olsztyn Economic Journal2019, No.14 (2), p. 179-194, DOI:10.31648/oej.3970.

Response: In this new version we have incorporated the information into the discussion. We believe that the information provided has contributed to improve the discussion. Please, see the third, fourth, fifth, sixth, seventh, and eighth paragraphs of the discussion section, pages 8 and 9.

  1. The Conclusions section should be expanded. Currently, it practically does not exist. The authors limit themselves to just two sentences: „The coordination between the social fabric demands and political will were considered to be the facilitating factors for LFP. Furthermore, in KI opinion, the presence of health and sustainability issues on the public agenda, the existence of a structured productive fabric and political changes represent an opportunity to implement LFP.” (line 303-306). It is not enough. What are the conclusions for producers from these studies? For consumers? For local purchase supporters? For government representatives?

Response: Thank you for the comment. In this new version, we have incorporated the information into the conclusions. Besides, we have incorporated a paragraph with policy implications in the discussion. Please, see the conclusion section and the last paragraph of the discussion, page 9.

  1. Moreover, in the Conclusions section, it should be clearly stated about the limitations of the method used and the study itself, which is qualitative and not representative. In the end, indicate further directions for research. Will the Authors seek confirmation of the obtained results in quantitative research?

Response: As recommended by the reviewer, we have included in the study limitations that it is a qualitative study carried out with 14 key informants, which does not allow us to extrapolate the results. Also, the conclusions indicate further directions for research. Please, see the seventh paragraph of the discussion, and the last paragraph of the conclusion, page 9.

  1. It would be better to include a research tool in the appendix.

Response: The data collection instrument is in Spanish. Taking into account the time to carry out the review, we have not sent it as an appendix. However, if you consider it necessary, we will proceed to translate the document into English.

Round 2

Reviewer 1 Report

The authors have addressed all my comments. So, the manuscript in sow suitable for publication.

Author Response

English language and style are fine/minor spell check required

Response: We appreciated the time you spent in reviewing this work. The spelling has been checked and corrected by an English native speaker.

Reviewer 2 Report

The authors made all changes suggested. The manuscript can be accepted.  

Author Response

(The authors gave the same response as above.)

Reviewer 3 Report

I'd like to thank the Authors for addressing the comments provided. Not all the amendments met my expectations but the quality of the manuscript has improved. The literature review is still poor.

Author Response

English language and style are fine/minor spell check required

Response: We appreciated your time in reviewing this work. The spelling has been checked and corrected by an English native speaker.

I'd like to thank the Authors for addressing the comments provided. Not all the amendments met my expectations but the quality of the manuscript has improved. The literature review is still poor.

 Response: in response to the comments, in this new version we have included new references in the introduction and in the discussion sections.

We have included the following references:

  1. Maciejewski, G. Food consumption in the Visegrad Group Countries–towards a healthy diet model. Studia Ekonomiczne 2018, 361, 20-32.
  2. Shrestha, R.M.; Schreinemachers, P.; Nyangmi, M.G.; Sah, M.; Phuong, J.; Manandhar, S.; Yang, R.Y. Home-grown school feeding: assessment of a pilot program in Nepal. BMC public health 2020, 20, 28, doi:10.1186/s12889-019-8143-9.
  3. Ni Mhurchu, C.; Vandevijvere, S.; Waterlander, W.; Thornton, L.E.; Kelly, B.; Cameron, A.J.; Snowdon, W.; Swinburn, B. Monitoring the availability of healthy and unhealthy foods and non-alcoholic beverages in community and consumer retail food environments globally. Obesity reviews 2013, 14 Suppl 1, 108-119, doi:10.1111/obr.12080.
  4. Mazarello Paes, V.; Ong, K.K. Factors influencing obesogenic dietary intake in young children (0-6 years): systematic review of qualitative evidence. BMJ open 2015, 5, e007396, doi:10.1136/bmjopen-2014-007396.
  5. de Jager, I.; Giller, K.E.; Brouwer, I.D. Food and nutrient gaps in rural Northern Ghana: Does production of smallholder farming households support adoption of food-based dietary guidelines? PLoS One 2018, 13, e0204014, doi:10.1371/journal.pone.0204014.
  6. Mason-D'Croz, D.; Bogard, J.R.; Sulser, T.B.; Cenacchi, N.; Dunston, S.; Herrero, M.; Wiebe, K. Gaps between fruit and vegetable production, demand, and recommended consumption at global and national levels: an integrated modelling study. The Lancet. Planetary health 2019, 3, e318-e329, doi:10.1016/s2542-5196(19)30095-6.

Please, see the introduction and the discussion section.